# Facile Synthesis of Functionalised Hyperbranched Polymers for Application as Novel, Low Viscosity Lubricant Formulation Components

**DOI:** 10.3390/polym14183841

**Published:** 2022-09-14

**Authors:** Sophie R. Goodwin, Amy Stimpson, Richard Moon, Lauren Cowie, Najib Aragrag, Sorin V. Filip, Andrew G. Smith, Derek J. Irvine

**Affiliations:** 1Centre for Additive Manufacturing, Faculty of Engineering, University of Nottingham, University Park, Nottingham NG7 2RD, UK; 2bp Applied Sciences, Innovation & Engineering, Technology Centre, Pangbourne, London SW1Y 4PD, UK

**Keywords:** hyperbranched polymers, catalytic chain transfer, lubricants, low viscosity

## Abstract

A novel, previously unreported, method for synthesising hyperbranched (HB) materials is detailed. Their use as additives to produce lubricant formulations that exhibit enhanced levels of wear protection and improved low-temperature oil viscosity and flow is also reported. The lubricant formulations containing HB additives were found to exhibit both significantly lower viscosities and improved in-use film-forming properties than the current industry standard formulations. To achieve this, alkyl methacrylate oligomers (predominantly dimers and trimers) were synthesised using catalytic chain transfer polymerisation. These were then used as functional chain transfer agents (CTA) to control the polymerisation of divinyl benzene (DVB) monomers to generate highly soluble, high polydispersity HB polymers. The level of dimer/trimer purification applied was varied to define its influence on both these HB resultant structures and the resultant HB additives’ performance as a lubricant additive. It was shown that, while the DVB acted as the backbone of the HB, the base oil solubility of the additive was imparted by the presence of the alkyl chains included in the structure via the use of the oligomeric CTAs.

## 1. Introduction

Industrial lubricants have had to evolve to match the increasingly stringent demands of evolving transportation technologies. For example, requirements have extended from acting as simple wear reduction and heat transfer media in cast-iron machinery. Now they are also expected to function as antioxidants and provide protective film coatings in modern combustion engines [1,2]. Two common components of current industrial lubricants are viscosity and friction modifiers, which increase operational efficiency and reduce wear, respectively [3]. Friction modifiers (FM) reduce energy losses generated by internal friction and maintain boundary films to prevent wear [4]. In practice, whilst the base oil gives some boundary friction reduction, it is not regarded as a true FM [5]. Rather, additional additives, including polymers, are used to reduce friction [6]. However, there are concerns related to polymeric additive shear stability in this context [7]. Traditionally, these additives are amphiphilic molecules, i.e., they contain both a straight-chain hydrocarbon “tail” group and a polar “head” group, such as amines, amides, carboxylic acids and phosphoric/phosphonic acids. The latter is proposed to interact with the metal surfaces within the engine to anchor the materials [8,9,10]. Alternatives to polymeric additives are inorganic additives, such as molybdenum disulphide (MoS_2_), which has a crystal structure. These deposit on the metal surface, and allow sliding and shearing to occur [3,11]. More recently, the use of MoS_2_ nanoparticles has been found to give ultralow friction coefficients [12]. 

Viscosity modifiers (VMs) are often high molecular weight, linear polymers added to improve the viscosity index of the oil. This allows the oil to operate efficiently over a wider temperature range. These multi-grade oils enable easier cold cranking and starting, which delivers improved fuel efficiency. Commonly used VMs are linear olefin copolymers, such as ethylene-propylene, hydrogenated styrene-diene and poly(alkyl)methacrylate copolymers [3]. However, these suffer from strong, attractive and intermolecular/chain interactions, which limit the minimum viscosity that can be achieved at low temperatures. These interactions have been shown to be reduced if nonlinear, 3-dimensional (3D) polymer structures are used [13]. 

Of these, hyperbranched (HB) and dendrimeric polymers have been shown to exhibit the greatest difference from linear materials, in terms of viscosity build when they are added to liquid media [14]. The high level of branching is hypothesised to impart what is referred to as a “globular” molecular structure that limits interchain contact. Dendrimers are extremely branched, well-defined (monodisperse), 3D macromolecules, typically synthesised via highly controlled, multistage, sequential and iterative reactions [15,16,17]. Their “porous” structures emanate from a central core and have been shown to be ideally suited for encapsulating small molecules. Consequently, they have been shown to transport molecules for fragrance distribution, drug delivery, diagnostics, catalysis and light-emission applications [18,19,20,21]. Their material properties, which include altered polymer melt-flow characteristics, have also garnered interest [22,23]. However, research into their potential end-use applications has been hampered by the labour, time and cost-intensive routes required for their production [16,24,25]. 

By contrast, HB polymers can be more rapidly (hours rather than weeks), cheaply (single, not multistep) and flexibly (larger range of applicable reagent monomers) synthesised than dendrimers [26,27,28,29,30]. Producing structures similar to dendrimers but less well-defined, with the potential to deliver the same performance at a much-reduced cost [26]. This has led to an interest in them running from petrochemical applications, such as in this study, through to glycopolymers for use in biomedical applications [9,31,32]. For example, they cannot efficiently pack and so exhibit intrinsically low viscosities, while also possessing the high molecular weights common to viscosity modifiers [24,33,34]. Recently, there have been a number of examples of branched polymers being used to modify viscosities in lubricant and heavy oil applications, including transportation [35,36,37,38,39]. However, polymer addition has been reported to lead to a reduction in oil/lubricant quality [40,41]. In this respect, the importance of the oil solubility of these polymers has been shown to be important in minimising these issues and also improving traditional additives [42]. Thus, characterisation of the oils/formulations components has aided the development of polymeric viscosity modifiers by allowing the understanding of what intermolecular interactions may be promoted upon the introduction of an additive [43,44,45].

Furthermore, in specific relation to this study, as difunctional monomers are utilised to synthesise HB polymers, care must be taken to avoid cross-linking. This leads to the formation of insoluble gels, rather than the desired solvent-soluble HB materials [26]. This coupling can occur in conventional free radical polymerisation at very low difunctional monomer concentrations (<<10%) and at low monomer conversions (<20%) [46]. 

Catalytic chain transfer polymerisation (CCTP) is an industrially viable polymerisation control strategy which typically uses a cobalt organometallic complex as a chain transfer agent (CTA). The CTA is catalytic in nature, thus, very low-molecular-weight polymers (e.g., dimer/trimers) can be produced using only parts per million quantities of CTA with some monomers [47]. Recently, researchers including Guan, Sherrington, Haddleton and Irvine all reported the use of catalytic chain transfer polymerisation (CCTP) to synthesise HB polymers [46,48,49,50]. With difunctional monomers, CCTP prolongs HB gelation within a range of monomer types, provided that either (a) the branching monomer concentration is kept low (<5 wt%) [46] or (b) the level of overall conversion is limited to specific levels [48,49,50,51,52]. Additionally, CCTP-terminated methacrylate chains are terminated by a vinyl group, resulting in increased concentrations of vinyl chain ends when compared to other CTAs. This presents the possibility of further post-modification/polymerisation [53]. Furthermore, the low molecular weight CCTP oligomers of monofunctional methacrylates have also been demonstrated to act as free radical CTAs via a beta-scission mechanism. This leads to a fragment of the CTA becoming an intrinsic part of the product HB structure [53]. Importantly, for lubricant applications, this will allow for the introduction of functionality that will promote base-oil solubility.

This study investigated the synthesis and use of HB materials as viscosity/friction modifiers to evaluate if they deliver high levels of wear protection and improved low-temperature oil fluidity. Furthermore, the work reported in this manuscript conducts polymerisations mediated by species which are added to interact and influence the propagation of the chain end radical [54]. In fact, the HB polymers were synthesised via a previously unreported route, where lauryl methacrylate (LMA) dimers were produced using CCTP and then used as CTAs to control the polymerisation of DVB. This method generated high conversions of highly polydisperse, base oil soluble HB polymers via the inclusion of the lauryl moiety in the HB structure. Solvent/oil soluble hyperbranched divinylbenzene (hbDVB) was of interest as a friction modifier because it can (a) be delivered as part of a liquid formulation and (b) cross-link *in situ* upon hot engine surfaces to produce protective films [55]. These HB polymers were then subjected to tribological testing and their performance was compared to industry-standard ethylene-propylene linear copolymers. 

## 2. Results and Discussion

### 2.1. Synthesis of Oil Soluble Hyperbranched Polymers

Hyperbranched divinylbenzene (hbDVB) homopolymers are of interest as friction modifiers due to their ability to exhibit oil solubility, so that they can be delivered as part of a liquid lubricant formulation. They also exhibit the potential to cross-link *in situ* on hot engine surfaces to produce protective films [55]. For this study, the “oil” of interest was poly(alpha-olefin) base oil 04 (PAO4), a hydrocarbon base oil used in the final lubricant formulations. However, an initial inspection of hbDVB solubility showed that it was essentially insoluble in PA04, this was attributed to the aromatic nature of DVB. Hence, the DVB monomer (Figure 1a) was initially directly copolymerised with a range of mono-olefin acrylate/methacrylate monomers (structures shown in Figure 1b–e). These contained long pendant alkyl chains to improve the base oil solubility of the subsequent HB copolymers. 

It was found that by using a ratio of either 1:9 or 2:8 (*v/v*) DVB:mono-olefin monomer when conducted in the presence of either bis[(difluoroboryl)diphenylglyoximato]cobalt(II) (PhCoBF) (acrylate) or 2,2,6,6-tetramethylpiperidin-1-yl)oxyl (TEMPO) (methacrylate), oil soluble copolymers were produced with identifiable gel points. These direct copolymers were proposed to be statistical copolymers, with a purely random mix of DVB and a monofunctional monomer. A representation of this copolymer structure is shown in Figure 2a. 

The copolymers synthesised from DVB and lauryl methacrylate/acrylate (Figure 1b,c) were chosen for further study. This was because they demonstrated the greatest level of base oil solubility (fully dissolving in PAO4 at 50 °C, Table 1) and exhibited the most straight-forward processing conditions, as shown in Figure 1c–e, in copolymers due to their physical form.

A key reason behind the processing advantages with LMA and LA was their low melting points of −7 °C and 4 °C, respectively, which reduced mixing issues with the DVB. By comparison, the melting points of SMA (18–20 °C, Figure 1d) and SA (32–34 °C, Figure 1e) lead to very lengthy mixing times to achieve homogeneous mixtures. However, the direct CCTP mediated polymerisation with DVB at both the 1:9 and 2:8 ratios was found to be inefficient. Thus, only relatively low levels of LMA could be incorporated into this random structure, which limited the achievable solubility of the final copolymer. Thus, to improve both the efficiency of copolymerisation and produce polymers with higher alkyl content, an alternative copolymerisation strategy was adopted. The new method centres on producing low molecular weight oligomers of LMA from the pure alkyl monomer by employing a high concentration of the CCTP catalyst, PhCoBF. In this way, a solution of monomer/dimer/trimer, with an oligomer yield of ~40% was achieved and confirmed by NMR/GPC analysis. LA was not used in this strategy because of the relative inefficiency of CCTP with acrylate monomers. These CCTP methacrylate-based dimer/trimer species can act as CTAs via a β-scission mechanism, so simultaneously control the polymerisation to produce hbDVB and introduce the required hydrocarbyl character [53]. 

These oligomer solutions were used in two methods to make HB polymers. To produce what were referred to as “semi-block-like” structures (Figure 2b), the CCTP oligomer solutions were polymerised with DVB in a ratio of 2:8, DVB:LMA oligomers (*v/v*), without any further purification, other than the removal of the residual PhCoBF catalyst. This ratio was chosen in an attempt to maximise the alkyl incorporation achieved. The term ‘semi-block-like’ refers to the use of the monomer/dimer/trimer solution as a CTA. Thus, some of the polymer end groups will be terminated with the CTA’s lauryl functionalised-olefin groups. However, as there will also be some LMA monomers present, some random polymerisation will be involved. Thus, the morphology of the HB polymer will also contain some random polymer characters. The ‘block-like’ structures (Figure 2c) were synthesised by reacting DVB with the same ratio (2:8) of a distilled solution of LMA CCTP oligomer products that reduced the monomer concentration from 60% to 19%. Consequently, the HB polymer had a higher probability of being terminated with the LMA CTA fragment, and reduced/no random character would be present. Subsequent application testing confirmed that both semi-block-like and block-like copolymers were soluble in the base oil at room temperature, indicating improved incorporation of LMA when using the oligomer CTAs. As a general procedure, all HB polymers were first synthesised until the gel point of the reaction was reached. Then, a repeat reaction was performed, which was quenched, and the polymer precipitated before the identified reaction gel time was reached. The results of all three reaction variants are summarised in Table 2, also an example of NMRs of the oligomer mediating agents and an example of an HB polymer have been included in the Appendix A. 

The crude polymer yield, i.e., the yield of precipitated polymer with no further purification, is reported to give an indication of the efficiency of the reaction. This is because all the polymers synthesised exhibited a glass transition (T_g_) below room temperature. Thus, the precipitated product could not be fully recovered from the antisolvent without the risk of crosslinking reactions between the individual HB polymers. To bypass that, we focused on concentrating these materials with precipitation and applying heat to drive off the residual solvent, which is a standard method for methacrylate polymers. However, it was found that with HBs, this increased the likelihood of the HB-to-HB links occurring. The result of which was the formation of a gelled 3D network, thus, work-up was conducted at room temperature. The potential secondary actions of this nature have been observed with other architectural polymer types (i.e., polymers with three-dimensional structures) [56,57]. From the data in Table 2, it is apparent that these reactions all produced a high yield of polymer compared to other literature-reported HB methods. Therefore, sufficient material could be generated for application testing [50,52,58,59,60,61]. A key initial observation was that these LMA oligomers produced materials had very high Đ values but were, in fact, still solvent/base oil soluble. This allowed them to both undergo GPC analysis and be used in further material testing (see Appendix A). The Đ value of a random polymer was observed to be the lowest (Table 2, entry 1), while the block-like polymers Đ values raised to ~50 and finally the semi-block-like typically had Đ values greater than 100. The lower M_w_ of the block-like structure compared to the semi-block-like was proposed to be due to the higher concentration of the CTA moieties in the distilled reagent mixture compared to the crude.

High Đ values are characteristics of HB polymers due to the nonlinear nature of their structure. Thus, they do not have a proportional relationship between hydrodynamic volume and molecular weight. Equally, HB polymers, due to their branched structures, do not exhibit the same relationship between hydrodynamic volume and molecular weight as linear materials. Thus, their passage through the GPC column will be altered, and so the predicted M_n_ and M_w_ values, which are based upon comparison to the commercially produced linear standards, will not be correct. However, these values were unexpectedly high, especially for soluble polymers. Thus, additional sets of experiments were prepared to identify if these Đ values were repeatedly achieved using this technique. These experiments involved the isolation of “kinetic” samples at 10-, 20- and 30-min points from a repeat reaction that was run through to the gelation point. Results from a reaction using purified LMA oligomers are shown as Entries 4–6 in Table 2. This data showed that the Đ did repeatably rise to the 50–70 region as the reaction reached approximately 30 min. Meanwhile, the random copolymers’ lower Đ value was attributed to the reaction being in the early stages of hyperbranching when it was quenched. This was due to greater reaction rate retardation caused by the high efficiency of the CCTP mediating agent. Its efficiency was also thought to result in very short interbranch distances. These factors associated with PhCoBF use were also thought to explain the lower polymer yield compared to the semi-block-like/block-like reactions. 

As a result of these materials being solvent/base oil soluble, they were passed on into application testing, although it is proposed that the actual value of the M_n_ and M_w_ molecular weights is relatively meaningless with such large Đ values. Rather, it was proposed that the differences in the polydispersities exhibited by the polymer mixtures may be a better “analytical value” by which to compare these materials. Ongoing work is trying to define alternative routes to quantify these types of large Đ materials.

### 2.2. Application Testing of Controlled Morphology Hyperbranched Polymers

#### 2.2.1. Viscosity 

The kinematic viscosities of the HB polymers detailed in Table 2 dissolved in base oil at concentrations of 0.0, 0.25, 0.5, 1.0 and 5.0 wt% were recorded at both 40 °C and 100 °C. This data is shown in Figure 3, and from an inspection of the data, it was apparent that the viscosity for each solution increased as the concentration of each polymer increased.

However, the viscosity of the standard linear polymer increased at a far greater rate than that of the HB polymers. This was attributed to the branched/globular structure of the HB polymer, which means it is more tightly compact. Additionally, it contains relatively short interbranch and branch-to-terminal group-chain lengths. Therefore, it was hypothesised that this resulted in a lower hydrodynamic volume for the HB, when compared to the linear polymer. Thus, it is less easily penetrated by other chains, reducing intercalation/reptation and so producing a lower increase in viscosity as the concentration of the polymer increases. 

It was also clear from Figure 3 that the trend in viscosity increased when comparing the HB variations reflected by the molecular weight of the polymers. The highest molecular weight semi-block-like structure exhibited a higher viscosity compared to the lower molecular weight block-like species at the same concentration. However, even the highest molecular weight HB, when included in the formulation at a 5 wt%, produced a significantly lower viscosity increase than the linear benchmark polymer at one fifth of this concentration.

#### 2.2.2. Film Formation

The copolymer solutions in base oil were investigated to determine their ability to form surface films created from the included HB additives. It was hypothesised that the high functional group densities exhibited by the HB additives would promote cross-linking in situ, and lead to the deposition of a strong, resistant film. This analysis was conducted using a mini-traction machine (MTM), where interferometry images were taken both before and after Stribeck curves were measured at 60, 90 and 120 °C. This comparison would define the persistence of the film, i.e., both its resistance to wear and its adhesion to the surface when subjected to the MTM conditions. In this method, the film thickness/characteristics of the test material were measured both by; (a) interferometry and (b) review of the level of the wear that resulted from the interaction of the MTM’s two surfaces. Figure 4 shows the interferometry images that result from the contact between the MTM’s ball and disc in pure base oil. 

The zero-point image (Figure 4A) shows the contact at 60 °C, before rolling, where the contact is perfectly round, with no deformation. Figure 4B shows where the ball was rolled at 100 mm s^−1^ for 1 h 15 min. Then Figure 4C shows the contact after the Stribeck curve was measured, which involved varying the rolling speed between 10 and 3200 mm s^−1^_._ Both Figure 4B,C exhibit elongation of the contact point, which is indicative of wear having occurred between the two surfaces. The level of wear observed was found to increase further as the temperature was increased to 90 °C (Figure 4D), such that the test was abandoned (Figure 4E) to prevent damaging the equipment. It is important to note that the interferometry image was still bright, indicating that no surface film had been produced with the base oil alone. Figure 5 contains the film thickness data for the pure oil formulation, presenting a 2D scatter plot (top) and a 3D surface plot (bottom) of the film thickness for each step. 

These visualisations highlight how, in Figure 5B, a film of significant thickness had initially formed, with some regions having an excess thickness of 150 nm. However, this film is not homogeneous, with large regions thinner than 10 nm, which was especially visible in the 3D surface plot. Following the 60 °C Stribeck curve measurement (i.e., Figure 5C), the overall maximum thickness of the film did not significantly change from the previous image. However, an increased proportion of the film was now less than 10 nm thick, suggesting that varying the rolling speed had worn away sections of the film. Furthermore, when the test temperature was increased to 90 °C (Figure 5D), the film thickness decreased significantly across the whole image. The majority of the film was now less than 10 nm thick, which suggested that an increasing temperature resulted in a significant reduction in film thickness. This was attributed to a decrease in the oil viscosity. This pure base oil solution data is summarised for each image in Table 3, also it suggests that the effect of film thickness increasing is more pronounced than that of varying rolling speeds. 

This conclusion was supported by the graphical visualisation of the data provided in the Appendix A. The Stribeck measurement (rolling data) was performed between steps 3 and 5, (entry B and C), and after step 7 (entry D). The data showed that the average film thickness decreased from 71 nm at entry B to 54 nm at entry C during the experiment, due to the changing rolling speed. However, the decrease in average thickness between entry C and D of ~50 nm corresponds to an increase in the test temperature from 60 to 90 °C. This is considerably greater than that caused by the Stribeck measurement. 

Meanwhile, the interferometry results of the comparative experiment, which utilised a 1 wt% solution of the semi-block-like HB polymer in oil, are shown in Figure 6. 

With this formulation, measurements could be conducted at a temperature of 120 °C (Figure 6F,G) without having to abandon the test due to excessive wear. Comparing the semi-block-like polymer solution data to that of the pure base oil, it was also apparent that there was very little wear occurring between the surfaces. This was shown by the lack of elongation of the contact point. The interferometry images also got steadily darker as the different rotation and temperature regimes were performed. This was indicative of the formation of a tribological film at the contact point between the surfaces and this film has been visualised in Figure 7.

Both the 2D scatter plot (top) and the 3D surface plot (bottom) indicated that a significant film formed rapidly, with the majority of the measured film having a thickness of >200 nm in plot Figure 7B. From this step, the film thickness continued to grow and become more uniform as the test progressed, with very little variation in film thickness seen after plot F. This indicated that varying rolling speed and increasing temperature both caused an increase in the thickness of the film formed, when the semi-block-like HB polymer was incorporated into the formulation. The film thickness summary for this test is shown in Table 4 and visualised in Appendix A, which corroborates that the observed formation of a protective, robust film was observed.

Initially, at step 3 (entry B), the average film thickness was 180 nm. The film then grew to a peak average thickness of 202 nm at step 13 (entry G) at 120 °C. Furthermore, at this point, the calculated errors (standard deviations) reduced significantly from ~35 to ~3 nm, indicating little variation in the film thickness across the wear scar. Comparing these results to the pure base oil data in Table 3, it was noted that the trend of decreasing film thickness was reversed. With the base oil formulation, the surface contact increased due to the loss of the film at the contact point. Thus, this was the root cause of the wear for the pure oil case. Rather, with the semi-block-like formulation, the thin film was found to grow with increasing temperatures and rotation regimes (i.e., after Stribeck measurements). This demonstrated that the addition of this HB copolymer resulted in the formation of a thin surface film, which was not worn down with an increasing temperature and applied force between the two surfaces. Currently, the range of the errors suggests that this can only be regarded as an indicative trend, but clearly shows differentiated performance from the base oil alone. 

Finally, the interferometry data for a 1 wt% solution in oil of the block-like HB polymer were collected and analysed using the methods described above. This data is shown in Appendix A in the ESI document. The interferometry images in Appendix A again show a lack of elongation, indicating that there is very little wear occurring between the surfaces. This also showed the significant formation of a surface film, as indicated by the darkening of the contact point. The visualisation data for the film are contained in Appendix A, along with additional interpretation comments. Thus, it was concluded that, in a similar manner to the semi-block-like polymer, a significant film forms rapidly, with the majority of the measured film having a thickness of >200 nm in plot 9B. From this step, the film thickness can be seen to continue to grow minimally and become more uniform as the test progresses, with very little variation in film thickness seen after plot 9D. This suggests that the block-like HB polymer forms a more robust film more rapidly than the semi-block-like copolymer. Furthermore, the film stabilises earlier in the test and is not affected as greatly by changes in temperature and rotation regime. The block-like polymer data summary is shown in Table 5 and visualised in Appendix A. 

The average thin film formed for the block-like HB solution was initially thicker than that of the semi-block-like polymer, with an initial film thickness at step 3 (entry B) of 195 nm, compared to 180 nm, respectively. Similar to the semi-block-like case, it was not degraded with increasing temperature and time, and grew as the experiment occurred, forming a maximum film thickness of 204 nm at step 11 (F). This is very similar to the 202 nm maximum average thickness observed for the semi-block-like HB, although the overall difference in thickness of 9 nm is not significant compared to the associated errors. However, the errors associated with the block-like system became very small in step 7 (D), indicating that a uniform film developed more rapidly than for the semi-block-like HB. The latter did not become uniform until step 11 (F). Once 90 °C was attained, the average film thickness was very similar for the two HB formulations. The more rapid block-like polymer film formation may be due to its smaller molecular size, allowing both quicker diffusion to the surface and/or better surface packing. 

The Stribeck curves for each test formulation, which were produced by the MTM apparatus, were then inspected. The results for the pure base oil are shown in Appendix A, as the 120 °C measurements could not be performed to prevent damage to the analytical device. The observed trend was an increase in the dimensionless friction coefficient as the speed of rotation decreased. This is because, at slower speeds, the surfaces are less able to entrain the test material; rather, it is forced out of the intersurface space due to the force applied between the surfaces. At 60 °C, the friction coefficient fluctuated considerably, but did exhibit an increasing trend, with a maximum coefficient of 0.38 being achieved. Meanwhile, at 90 °C, the friction rose to a maximum of 0.46, at which point the test was stopped to prevent apparatus damage. It was hypothesised that a decrease in the viscosity of the oil at higher temperatures resulted in lower entrainment of the oil, leading to greater wear. The influence this viscosity change has on the final performance of the HB-formulated systems will require further study. Thus, to better understand how it influences the mechanisms at each boundary, mixed or hydrodynamic lubrication was achieved. However, this more detailed study is outside the scope of this initial work programme and is the subject of an ongoing study. 

By comparing the data from the HB polymer formulations exhibited, a marked decrease in the friction coefficient is observed with the addition of both HB polymer types, see Figure 8. 

Even at low entrainment speeds, the HB systems outperform the pure base oil, e.g., at 60 °C, the semi-block-like HB results (Figure 8, left) peaked at a coefficient of friction of 0.14, compared to 0.38 for the oil without a polymer. Similarly, the 120 °C Stribeck curve results in an increased friction between the two surfaces, as would be expected. However, for the semi-block-like polymer, the expected trend of increasing friction coefficient with increasing temperature is reversed at 90 °C, where the friction decreases with increasing temperature. By cross-referencing this observation to the film thickness data in Table 4, this decrease in friction is accompanied by a significant increase in the average film thickness of ~16 nm at this point, compared to the average 60 °C value. This is the largest increase recorded during these measurements, with the typical film thickness increases being ~3 nm between tests. Thus, this 90-degree C result was attributed to the influence of the processes involved. Thus, the observation that is key to this work programme is that the addition of HB polymer results in the formation of persistent films within the test apparatus. Further, more detailed work is underway to define how this polymer film formation influences the lubrication regimes.

This need for further detailed studies of the lubrication regimes was further exemplified when these tests were performed using the block-like polymer in oil solutions. In this case, the expected trend of increased friction with increasing temperature was observed at all temperatures, Figure 8 (right). The block-like polymer formulation also produced lower friction than the pure base oil. Interestingly, it was noted that the friction coefficient was higher at each of the temperatures measured for the block-like polymer system when compared to the semi-block-like material. This was despite a thicker film having been formed for the block-like solution at lower temperatures (comparing the film thickness for the block-like and semi-block-like polymers at 60 °C of 195 nm and 180 nm, respectively). However, the block-like polymer solution was observed to undergo a lower average increase in film thickness (6 nm) than the semi-block-like polymer (12 nm) as the test proceeded (Table 4 and Table 5, respectively). Therefore, the increase in average film thickness does not produce the same decrease in friction as observed with the semi-block-like materials. Thus, the delivery of the lubrication effects is clearly related to more than just persistent film formation, and this more detailed study is currently underway.

As such, some initial results from additional tribological testing in the elastohydrodynamic lubrication regime are included in the Appendix A.

## 3. Conclusions

In this work, we have successfully synthesised and exemplified the enhanced performance of a new class of polymeric lubricant additives. These materials have been shown to provide enhanced wear protection and improved low-temperature viscosity/flow when compared to the current commercial benchmark polymeric additives. To achieve this, a novel method of chain transfer control over the polymerisation of DVB, using LMA oligomers, was developed. This method allowed the synthesis of HB polymers with very high Đ values, which were soluble in both solvent and PA04 base oil. Two variations were synthesised, the first using a crude oligomeric LMA mixture (termed semi-block-like), the second using a purified dimer/trimer LMA (block-like). Both HB polymer variants demonstrated excellent low viscosity lubrication and wear prevention properties. Specifically, the HB polymers demonstrated low kinematic viscosities (KV), good film formation, wear prevention and friction reduction compared to current standard linear additives. The hypothesis drawn was that the low KV was attributed to the “globular” structure of the HB polymers, while the film formation properties were attributed to the crosslinking of the high density of vinyl groups present in the HB at high temperatures and rolling incurred in the testing regimes. The films formed were not worn away with increasing temperature and rolling speed, and the wear scar produced did not show lengthening, indicating low levels of wear. The formation of the protective film also resulted in reduced friction between surfaces across a range of temperatures and rolling speeds. Overall, the extra time and cost of the purification step required to generate a block-like structure were not felt to be justifiable based on performance differences, so in future work, the semi-block-like polymer would be the preferred candidate for scaling up.

## 4. Experimental Methods

*Materials*: Unless otherwise stated, all reagents were used as received and without further purification, and all procedures were conducted under an inert argon atmosphere using standard Schlenk line techniques. Divinylbenzene 80% (DVB-80, technical grade, 80% difunctional monomer (*m-* and *p-*DVB), 20% monofunctional monomer (3- and 4- ethylstyrene)), lauryl methacrylate (LMA, 96%, 500 ppm MEHQ as inhibitor), lauryl acrylate (LA, technical grade 90%, 60–100 ppm monomethyl ether hydroquinone as inhibitor), stearyl methacrylate (SMA, mixture of stearyl and cetyl methacrylates, containing MEHQ as inhibitor), stearyl acrylate (SA, 97%, contains 200 ppm monomethyl ether hydroquinone as inhibitor), cyclohexanone (99%+), deuterated chloroform (99.8%+), tetrahydrofuran (THF, anhydrous, 99%+) and 2,2′-azobis(isobutyronitrile) (AIBN, 98%+) were all purchased from Sigma-Aldrich, UK. The commercially purchased PAO4 base oil was supplied by bp Applied Sciences, Pangbourne, UK and used as received. High-purity argon was purchased from BOC gases, UK. The externally purchased mediating agents bis[(difluoroboryl)diphenylglyoximato]cobalt(II) (CoPhBF) was obtained from DuPont and 2,2,6,6-tetramethylpiperidin-1-yl)oxyl (TEMPO, 98%) was purchased from Sigma-Aldrich, UK. Cyclohexanone solvent was purchased from Fischer, UK. Toluene solvent was obtained directly from the chemistry stores.

*Synthetic Procedures - Note on Gelation*: Gelation was defined as the point at which the solution ceased to be a free-flowing, easily stirred liquid and became a rubbery/solidified gel. This point was a readily observable change in physical form and apparent through visual inspection. However, in all cases, gelation was confirmed by withdrawing a small sample of the reaction mixture and attempting to dilute it in chloroform. Where gelation had occurred, the resultant sample presented as non-redissolvable material in the resulting solution. The time at which this occurred was defined as the gelation time and represented a change from an oligomeric/hyperbranched system to an extended, 3D, cross-linked system.

*Direct Nitroxide Mediated (NMP) and CCTP Mediated Copolymerisation of DVB and Lauryl/Stearyl (Meth)acrylate:* Copolymers were produced via the CCTP mediated reaction of DVB (2.0 mL, 1.828 g, 14.0 mmol) or by either LMA, LA, SMA or SA (8 mL, 27.3 mmol, 29.4 mmol, 20.4 mmol, 19.7 mmol). The desired quantities of these monomers, cyclohexanone (10 mL, 9.48 g, 96.6 mmol) and mediating agents were introduced into a Schlenk flask equipped with a magnetic stirrer bar and containing an inert argon atmosphere. No initiator was included in the reagent formulation because both styrene and DVB are known to self-initiate at reaction temperatures of approximately 150 °C. Thus, the reaction was allowed to self- initiate to produce the HBs, as has previously been reported by the authors [37]. For the methacrylate species 2,2,6,6-(tetramethylpiperidin-1-yl)oxyl (TEMPO, 46 mg, 0.294 mmol) was used as the polymerisation mediating agent, and for the acrylate species, bis[(difluoroboryl)diphenylglyoximato]cobalt(II) (PhCoBF, 26 mg, 0.0401 mmol). For both monomer families, the reaction vessel was then immersed in a preheated oil bath which was thermostatically controlled to remain at 150 °C. The initial reactions were run until gelation occurred, at which point the time was noted, and a repeat reaction was conducted. This repeat reaction was quenched 15 min before the time gelation was known to occur in the initial scouting experiment. Once cooled, the solution was added dropwise to cold (0 °C) methanol. The resulting precipitation was collected via filtration to provide a colourless (NMP experiments) or brown/white, viscous liquid (CCTP experiments colouration due to PhCoBF), which was then dried to constant mass under vacuum.

*Synthetic Procedures: Preparation of Crude Lauryl Methacrylate (LMA) Oligomers:* A solution of mixed LMA oligomers was synthesised via CCTP of LMA monomer. The degassed LMA monomer (100.0 mL, 86.8 g, 341.2 mmol), toluene (5.0 mL, 4.335 g, 47.2 mmol), PhCoBF (78.0 mg, 0.120 mmol) and AIBN (866 mg, 5.274 mmol) were introduced into a Schlenk flask equipped with a magnetic stirrer bar and containing an inert argon atmosphere. The reaction vessel was then immersed in a preheated oil bath, which was thermostatically controlled to remain at 80 °C. After 24 hrs, the vessel was removed from the oil bath and quenched to prevent further reaction, and an oligomer concentration of 40% was obtained.

Preparation of Purified Lauryl Methacrylate Oligomers.

A crude oligomeric solution of LMA was synthesised by the procedure outlined above: the crude product was then vacuum distilled to increase the oligomer concentration in the solution to 81%. The remaining 19% was lauryl methacrylate monomer.

*Hyperbranched Functional Polymer Synthesis* via *Copolymerisation of DVB and LMA Dimer:* Degassed DVB-80 (12.5 mL, 11.4 g, 87.8 mmol) and cyclohexanone (10 mL, 9.48 g, 96.6 mmol) were transferred to a Schlenk flask containing LMA oligomer (50 mL), PhCoBF (39.0 mg, 0.0601 mmol), a magnetic stirrer bar and containing an inert argon atmosphere. Again, no initiator in the reagent formulation was included as the reactions were allowed to self-initiate. The resulting solution was then heated rapidly to 150 °C by immersion in a preheated oil bath and held at that temperature for 35 min, after which it was immediately quenched to prevent gelation. Once cooled, the solution was added dropwise to cold (0 °C) methanol. The resulting precipitation that was formed via introducing the mixture to the low temperature solvent was collected via filtration at this reduced temperature. It was then warmed to room temperature to provide a brown/white, viscous liquid, which was then dried to constant mass. Yield = 63%. GPC: M_n_ = 37000, Đ = 102.7. The brown colouration was attributed to a small amount of retained cobalt catalyst. However, the precipitation solvent was highly coloured indicating that the majority of the PhCoBF has been removed via this separation process.

*Polymer Characterisation - Gel permeation chromatography (GPC):* This was performed using a refractive index (RI) detector with HPLC THF as the eluent. Analysis was performed at 40 °C with a flow rate of 1 mL min^−1^ through two PolarGel-M columns with a calibration range of 580−377 400 Da calibrated with 10 poly(styrene) narrow molecular weight distribution standards. All GPC equipment and standards were supplied by Polymer Laboratories (Varian). GPC data were analysed using the Cirrus GPC and Astra offline software packages.

*Nuclear Magnetic Resonance (NMR):*^1^H NMR spectra were obtained in CDCl_3_ on a Bruker AV3400 (400 MHz) spectrometer. Chemical shifts were referenced against residual solvent signal (^1^H = 7.26 ppm, ^13^C = 77.16 ppm) and processed using the MestReNova software. In this study NMR was primarily used for oligomer analysis as the resonances in the spectra of the HB materials were too broad to allow for additional structural information to be identified.

*Application Testing:* All tribological application testing was undertaken under the guidance of bp/Castrol at their facility in Pangbourne, UK. PAO4 was used as the control base oil medium in all tests. The tribological features of the polymers produced were tested under both hydrodynamic and boundary conditions to test both the viscosity of the solutions and film formation at surfaces.

*Base Oil Solubility Testing:* The solubility of different polymers produced in the above test methods was analysed through adding a small amount (20–100 mg) of the produced polymer into PAO4. The tests started at room temperature, and then with stirring, the temperature of the PAO4 was increased in 10 °C intervals.

*Kinematic Viscosity Measurements:* The kinematic viscosities (KV) of the synthesised materials were measured by passing concentrations of polymer (0.0, 0.25, 0.5, 1.0 and 5.0 wt%), dissolved in base oil, through a glass capillary viscometer. The time taken for a known volume of the solution to pass through the capillary was recorded. The tests were conducted according to the ASTM D2270 standard. The samples and the apparatus were prepared in accordance with the manufacturer’s recommendations printed in the user manual. The kinematic viscosity was recorded at both 40 °C and 100 °C

*Film Formation Measurements:* The ability of the polymer solutions in oil to form surface films was measured using a mini-traction machine (MTM). The MTM uses the frictional force between a rotating, highly polished spherical ball and an independently rotating highly polished glass disc. The ball is placed on the face of the disc in a reservoir of material to be measured, then the ball and disc are rotated to produce a rolling/sliding contact. The temperature of the reservoir can be increased, to allow for measurement at different temperatures. A force of 40 N was applied between the ball and the disc, which were then rotated at a speed of 100 mm s^−1^ for 1 h and 30 min with a slide/roll ratio (SRR) of 50%, to ensure both surfaces were covered with the oil solution. The speed of the rotation was then increased to 3200 mm s^−1^ and then reduced to 10 mm s^−1^ at 30 °C temperature intervals between 60–120 °C, while measuring the frictional coefficient to produce a Stribeck curve of frictional coefficient against speed of rotation. Interferometry images were taken before and after the Stribeck curve was measured to determine the thickness of the oil/polymer film on the contact.

*Film Formation Analysis:* The thickness (nm) of the film formed by the lubricating solutions was measured by interferometry at each coordinate over a circular grid of fixed radius, centred on the exact middle of the wear scar. This data was plotted in 2D and 3D for each step of the MTM measurement. For the 2D scatter plots, the data was binned to allow significant variations in the film thickness in each image to be observed more clearly. The mean, standard deviation, minimum and maximum thickness were calculated for each measurement.

## Figures and Tables

**Figure 1 polymers-14-03841-f001:**
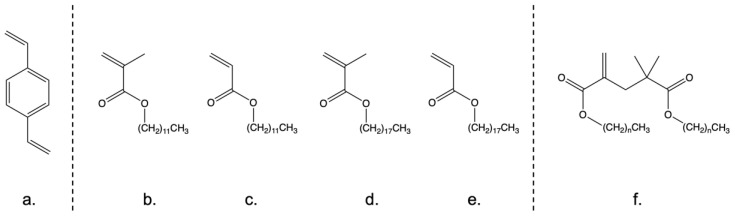
Structures of monomers and oligomer reagents used in synthesising functional HB polymers. Monomers = (**a**) DVB, (**b**) lauryl methacrylate, (**c**) lauryl acrylate, (**d**) stearyl methacrylate and (**e**) stearyl acrylate. Oligomers = (**f**) dimer created from lauryl/stearyl methacrylate, where n is 11 (lauryl) or 17 (stearyl).

**Figure 2 polymers-14-03841-f002:**
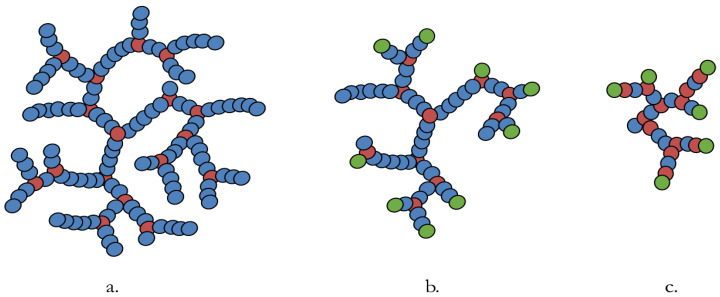
Structures of the (**a**) random, (**b**) semi-block-like and (**c**) block-like HB copolymers. The red spheres represent multifunctional monomers, blue spheres the monofunctional monomers and green spheres the macromeric CTA fragments.

**Figure 3 polymers-14-03841-f003:**
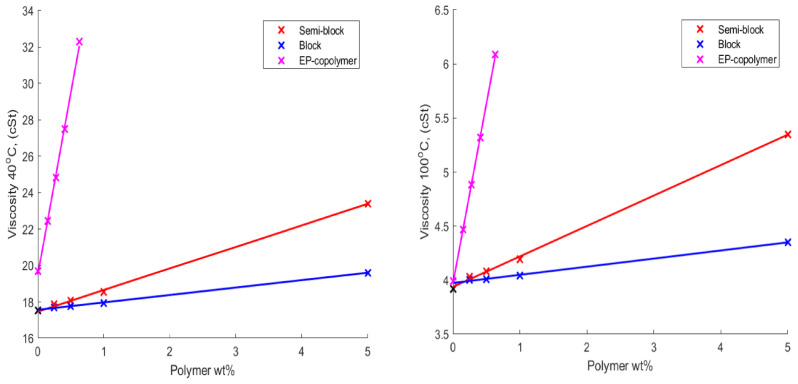
Plot of the change in viscosity of both linear and HB polymer base oil solutions with polymer content measured at 40 °C (**left**) and 100 °C (**right**), where the EP-copolymer is the current commercial linear polymer used in the industrial formulations.

**Figure 4 polymers-14-03841-f004:**
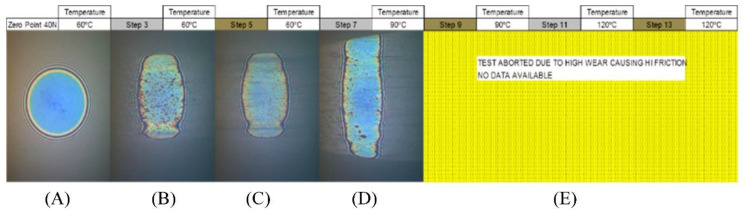
Interferometry images for an experiment using pure base oil only as the lubricant where image (**A**) is the zero-point image (i.e. prior to analysis) and images (**B**–**E**) depict different time point in the analytical process.

**Figure 5 polymers-14-03841-f005:**
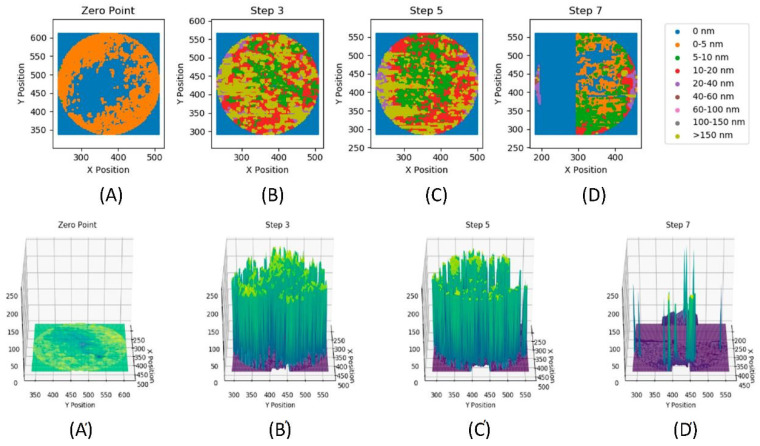
Plots of the film thickness (nm) for a pure oil formulation. (**A**–**D**) **Top**: two-dimensional scatter plots for each step, where the data has been binned, as indicated in the legend. (**A′**–**D′**) **Bottom**: three-dimensional surface plots for each step. The labels (**A**–**D**) correlate across the two plots and with the labels in Figure 4.

**Figure 6 polymers-14-03841-f006:**
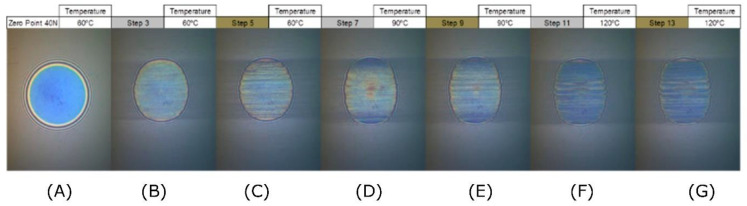
Interferometry images for an experiment utilising a 1 wt% semi-block-like HB polymer in an oil solution where image (**A**) is the zero-point image (i.e. prior to analysis) and images (**B**–**G**) depict different time points in the analysis.

**Figure 7 polymers-14-03841-f007:**
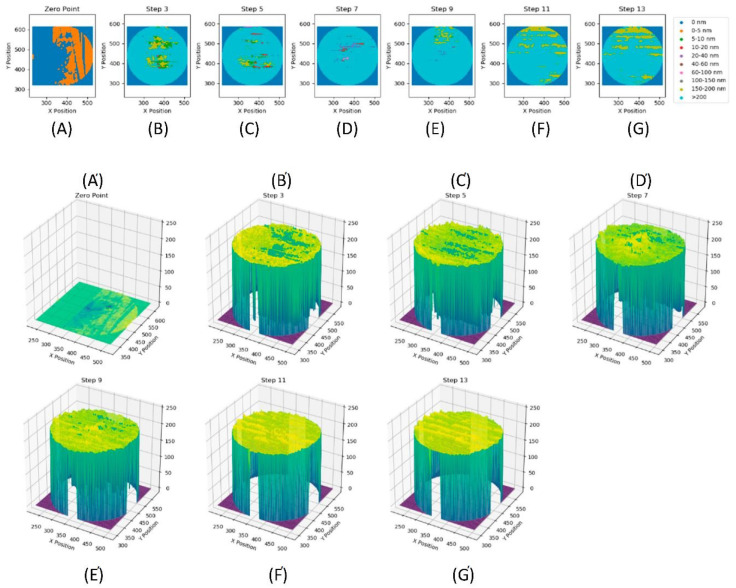
Plots of the film thickness (nm) for a 1 wt% semi-block-like HB polymer and base oil formulation. (**A**–**G**) **Top**: two-dimensional scatter plots for each step, where the data has been binned, as indicated in the legend. (**A′**–**G′**) **Bottom**: three-dimensional surface plots for each step. The labels (**A**–**G**) correlate across the two plots and with the labels in Figure 6.

**Figure 8 polymers-14-03841-f008:**
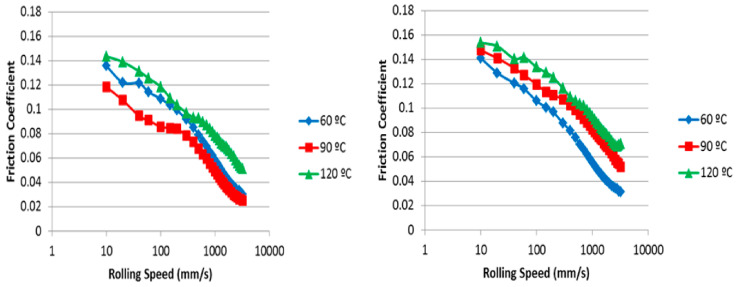
Stribeck curve from an experiment using a 1 wt% semi-block-like HB polymer in oil solution (**left**) and 1 wt% block-like HB polymer in oil solution (**right**) as the lubricant.

**Table 1 polymers-14-03841-t001:** Solvation properties of random structure copolymers prepared using a 2:8 ratio of alky reagent:DVB in base oil.

Comonomer	Solvation Temp (°C)	Appearance in Solution
SA	70	Colloidal Suspension
SMA	70	Clear Solution
LA ^a^	50	Clear Solution
LMA	50	Clear Solution

^a.^ Species was unable to be isolated from the reaction mixture.

**Table 2 polymers-14-03841-t002:** Yield and molecular weight data of the copolymers produced.

Entry	Structure	CTA Oligomer Used	Time (Min)	Yield ^a^ (%)	M_n_ ^b^ (g mol^−1^)	Đ ^b^
1	Random	PhCoBF	35	53	3650	4.9
2	Semi-Block-like	Crude LMA	35	66	3700	102.7
3	Block-like	Purified LMA	35	65	4400	58.4
4	Block like	Purified LMA	10	na ^c^	2500	5.5
5	Block like	Purified LMA	20	na ^c^	3300	6.7
6	Block-like	Purified LMA	30	na ^c^	4200	74.7
7	Block-like	Purified LMA	40	Gel	Gel	Gel

^a.^ determined gravimetrically; ^b.^ determined via GPC, ^c.^ na = not available—these results were from small kinetic samples taken at time points from a reaction that was allowed to run through to the gelation point. Thus, a representative gravimetric yield could not be obtained from these small samples.

**Table 3 polymers-14-03841-t003:** Summary of the film thickness data obtained from interferometry for a pure base oil solution. The blue shading indicates that the measurement was conducted after film buildup (i.e., following a period with no change in temperature or rolling speed). The orange shading indicates that the measurement was conducted after a Stribeck Curve measurement was conducted.

Step No	Maximum Thickness [nm]	Minimum Thickness [nm]	Mean Thickness [nm]	Standard Deviation [nm]	Measurement Temperature [°C]
**A (Zero Point)**	5.0	−10.0	0.3	1.5	60
**B (Step 3)**	241.0	−10.0	71.1	89.4	60
**C (Step 5)**	233.0	−6.0	54.0	83.0	60
**D (Step 7)**	240.0	−10.0	4.5	19.8	90

**Table 4 polymers-14-03841-t004:** Summary of the film thickness data obtained from interferometry for a 1 wt% semi-block-like HB polymer/base oil formulation. The blue shading indicates that the measurement was conducted after film buildup (i.e., following a period of constant temperature and rolling speed). The orange shading indicates that the measurement was conducted after a Stribeck Curve measurement was conducted. The plot (right) is a visual representation of this data (B–G).

Step No	Maximum Thickness [nm]	Minimum Thickness [nm]	Mean Thickness [nm]	Standard Deviation [nm]	Measurement Temperature [°C]
**A (Zero Point)**	4.0	−10.0	0.2	1.3	60
**B (Step 3)**	217.0	0.0	179.9	67.9	60
**C (Step 5)**	231.0	0.0	184.3	63.0	60
**D (Step 7)**	241.0	7.0	196.3	53.6	90
**E (Step 9)**	230.0	0.0	200.3	35.4	90
**F (Step 11)**	216.0	127.0	202.3	3.4	120
**G (Step13)**	219.0	74.0	202.3	4.2	120

**Table 5 polymers-14-03841-t005:** Summary of the film thickness data obtained from interferometry for a 1 wt% block-like HB polymer/base oil formulation. The blue shading indicates that the measurement was conducted after film buildup (i.e., following a period of constant temperature and rolling speed). The orange shading indicates that the measurement was conducted after a Stribeck curve measurement was conducted.

Step No	Maximum Thickness [nm]	Minimum Thickness [nm]	Mean Thickness [nm]	Standard Deviation [nm]	Measurement Temperature [°C]
**A (Zero Point)**	6.0	−10.0	0.7	1.7	60
**B (Step 3)**	218.0	−3.0	194.9	44.6	60
**C (Step 5)**	227.0	1.0	194.4	46.6	60
**D (Step 7)**	223.0	80.0	200.1	12.1	90
**E (Step 9)**	227.0	118.0	201.0	9.9	90
**F (Step 11)**	223.0	75.0	204.4	8.8	120
**G (Step13)**	220.0	118.0	201.4	13.3	120

## Data Availability

Due to confidentiality agreements with research collaborators, supporting data can only be made available to bona fide researchers subject to a nondisclosure agreement. Details of the data and how to request access are available at the University of Nottingham data repository.

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
