# Peer review of "Facile Synthesis of Functionalised Hyperbranched Polymers for Application as Novel, Low Viscosity Lubricant Formulation Components"

_polymers, 2022, doi:10.3390/polym14183841_

Round 1

Reviewer 1 Report

This is an interesting study and it is recommended to be published after some revision on the basis of comments below.

COMMENTS

1.

Scheme 1 is misleading, because it is known the bifunctional monomers do not polymerize completely, that is unreacted double bonds remain in such structures. This should be drawn correctly. The extent of unreacted double bonds influences strongly the properties of such polymers, especially in application at higher temperatures, because gelation can occur at such temperatures.

2.

The sample coding is hard to understand and follow. This should be corrected.

3.

The authors write that “It was found that by using a ratio of either 1:9 or 2:8 (v/v) DVB:mono-olefin monomer controlled with 2,2,6,6-Tetramethylpiperidin-1-yl)oxyl (TEMPO) (methacrylate) or bis[(difluoroboryl)diphenylglyoximato]cobalt(II) (PhCoBF) (acrylate), oil soluble copolymers were produced with identifiable gel points”. What the authors mean on “controlled”? Nothing is controlled here. The authors prepare soluble polymers by stopping the reaction before gelation occurs.

4.

In the Results and Discussion section, it is unclear which ratio is used 1:9 or 2:8 for the specific copolymerizations. This should be clearly provided in each case.

5.

Table 1 does not provide sufficient information. Is DVB the other comonomer? If so, what methacrylate/acrylate and DVB ratios were used.

6.

The authors write that “the Mn and Mw molecular weight is relatively meaningless with such very large Đ values polymers”. Presenting the GPC curves of the resulting oligomers (dimers, trimers) and the hyperbranched polymers should be definitely provided in the manuscript and not in the Supporting Information. In addition, detailed sample identification, that is, the conditions for their preparation should be provided for the correct and full information of the readers, that is the applied starting materials, comonomers, their ratios and reaction times.

7.

The NMR spectra of the oligomers should be provided. Additionally, the NMR spectra of the hyperbranched polymers would be also informative for the readers, especially in observing the relative quantities of the unreacted double bonds.

8.

The yields are missing for entries 4, 5 and 6 in Table 2. These should be provided.

9.

The authors write that “the precipitation product couldn’t be fully separated from the anti-solvent, without the risk of crosslinking the polymer collected”. What kind of crosslinking would occur? This should be explained by the authors in details.

10.

The authors write that “HB polymers don’t interact with the GPC columns in the same manner as a linear polymer due to both their branched structures”. What kind of “interaction” the authors have in mind. If a GPC column interacts with the polymer sample, then not only size exclusion but the mode of interaction also determines the shape of the elution curves.  

11.

The authors should provide detailed characterization on the EP-copolymer used for comparison.

12.

The authors provide comparison with the EP-copolymer only for viscosity but not for the film formation and the friction coefficient. Therefore, the advantage or disadvantage of the investigated hyperbranched polymers cannot be correctly judged.

13.

The authors write that “For the methacrylate species 2,2,6,6-Tetramethylpiperidin-1-yl)oxyl (TEMPO, 46 mg, 0.294 mmol) was used as the CTA”. TEMPO is not a chain transfer agent (CTA), but a radical trap (terminating agent) at relatively low temperatures and polymerization initiator with reversible carbon-oxygen bond scission at the applied polymerization condition at 150 C.

14.

The initiator and its quantity used in the TEMPO and PhCoBF mediated radical polymerizations in the presence of DVB is not listed in the Experimental.

Reviewer 2 Report

With a suitable title, an informative abstract and a rich number of experimental procedures sustaining a properly selected research design, the manuscript desereves publication, but before is a need to change/complete the following aspects:

a)The paper is clearly written but not well organized ( as example as can nominate a too long introduction  based on too old references )

b) In fact the chapter references has too many titles older than five years and is not able to sustain properly the novelty of the paper presenting  with  details a comparison of new group of additives and  the current commercial polymeric additives

Reviewer 3 Report

Title: Novel, Low Viscosity Lubricant Formulation Components via Facile Synthesis of Functionalised Hyperbranched Polymers: Formulation and Testing.

Recommendation: Major revisions needed as noted.

The manuscript is well written and has very attractive idea to produce But, there are a few questionable points that the authors should try to more clearly address.

1. Author should change the title of the manuscript; try to make it short and more meaningful.

2. The wording of the abstract is not good, try to make it more concise.  

3. The length of sentences are too large, which leads the readers to confusion… Therefore, authors are advised to write concise and short content for easy understanding. i.e ( 1. Hyperbranched divinylbenzene (hbDVB) homopolymers are of interest as friction modifiers due to their ability to exhibit oil solubility so that they can be delivered as part of liquid lubricant formulation, but they also exhibit the potential to cross-link in-situ upon the hot engine surfaces to produce protective films; 2. Hence, the DVB monomer (Figure 1a) was initially directly copolymerised with a range of mono-olefin acrylate/methacrylate monomers (structures shown in Figure 1 b-e) with long pendant alkyl chains in order to improve the solubility in the base oil of the subsequent hyperbranched copolymers; 3. The term ‘semi-block-like’ refers to the fact that, by using the monomer/dimer/tri-mer solution, some of the polymer end groups will be terminated with lauryl functional-ised olefin groups, but that there will also be some LMA monomer present which will be involved in random polymerisation. Thus, the morphology of the HB polymer will also contain some random polymer character; 4. The ‘block-like’ structures (Figure 2c) were synthesised by reacting DVB with a dis-tilled solution of the LMA CCTP oligomer products used to produce the semi-block-like structure in order to reduce the monomer concentration in the solution from 60 % to 19 %, meaning that the concentration of the dimer/trimer chain transfer agent was considerably higher.).

4. Carefully check the superscript and subscript as well as the “oC” throughout the manuscript. You should add space between the digit and the oC. i.e 25 oC.

5. Quality of Picture and Graph should be enhanced.

6. Style and format of the Graph should be same.

7. Synthesis of the Oil Soluble Hyperbranched Polymers is not clear. Try to explain it in detail.

8. Must need careful revision of the manuscript, for grammatical structural and typo mistakes.

Round 2

Reviewer 1 Report

This paper still needs additional revision on the basis of comments below.

COMMENTS

1.

The term ”control agent” for TEMPO is absolutely wrong. This compound reacts reversibly with radicals at high temperatures, and thus suppresses the concentration of radicals which are able either to take part in propagation with the monomer(s) or react with each other leading to termination either by recombination or disproportionation. Thus, the correct term is “TEMPO mediated radical polymerization”.  Neither TEMPO nor PhCoBF “control” the inter-branch distances as claimed by the authors. The average inter-branch distance is the function of the ratio of a given monomer and bifunctional (or multifunctional) monomer, plus the reactivity ratios between them. Therefore, cancelling the “control agent” expression in the whole manuscript is necessary.

The same holds for the term “controlled”.  This highly undefined term should also be deleted completely in the whole manuscript. The result of any polymerization depends on the relative rates of the elementary polymerization reactions. This is well documented by the multimodal GPC traces, that is, the multimodal molecular weight distribution in Figure S1 indicates the complexity, that is, the fully “uncontrolled” nature of the investigated polymerizations . In this context, the authors should study, refer to and cite the following publication in this manuscriptin relation to use the correct terminology:

Ivan, B. Terminology and classification of quasiliving polymerizations and ideal living polymerizations on the basis of the logic of elementary polymerization reactions, and comments on using the term “controlled”. Macromol. Chem. Phys. 2000, 201, 2621-2628.

2.

NMR spectra of the products, even with overlapped signals, are still missing from this manuscript. The authors are encouraged to display the NMR spectra of the dimers and the hyperbranched polymers at least in the Supporting Information.

3.

On page 5 at the bottom, the authors describe the gelation of methacrylate type hyperbranched polymers. In relation to this problem, the authors are recommended to study, refer to and cite the following publication which applies post-polymerization derivatization for methacrylate polymers, and investigating such polymers by the authors  might also be of interest:  

Szanka, A.; Szarka, G.; Ivan, B. Multi-methacrylated star-shaped, photocurable poly(methyl methacrylate) macromonomers via quasiliving ATRP with suppressed curing shrinkage. Polymer 2013, 54, 6073-6077; Szanka, A.; Szarka, G.; Ivan, B. Poly(methyl methacrylate-co-2-hydroxyethyl methacrylate) four-arm star functional copolymers by quasiliving ATRP: equivalent synthetic routes by protected and nonprotected HEMA comonomers. Journal of Macromolecular Science, Part A 2014, 51, 125-133.

4.

In the caption of Table 1, copolymer should be written without hyphenation (see IUPAC Nomenclature rules).

Author Response

See comments in attachment

Reviewer 2 Report

The revised manuscript is a better paper and in my opinion despite the fact that data organization  Is not the best, the manuscript could be published in present form

Author Response

No further comments found to respond to.

Reviewer 3 Report

Now they have added more information. 

Author Response

(The authors gave the same response as above.)
